# Detection of Potato Pathogen *Clavibacter sepedonicus* by CRISPR/Cas13a Analysis of NASBA Amplicons

**DOI:** 10.3390/ijms252212218

**Published:** 2024-11-14

**Authors:** Svetlana A. Khmeleva, Leonid K. Kurbatov, Konstantin G. Ptitsyn, Olga S. Timoshenko, Darya D. Morozova, Elena V. Suprun, Sergey P. Radko, Andrey V. Lisitsa

**Affiliations:** 1V.N. Orekhovich Institute of Biomedical Chemistry, 10 Pogodinskaya St., 119121 Moscow, Russia; diny1204@yandex.ru (S.A.K.); leonid15@mail.ru (L.K.K.); konstantin157@yandex.ru (K.G.P.); ryzhakova.olga@list.ru (O.S.T.); darya.d.morozova@gmail.com (D.D.M.); lisitsa052@gmail.com (A.V.L.); 2The Chemistry Faculty of the M.V. Lomonosov Moscow State University, 1/3 Lenin Hills, 119991 Moscow, Russia; lenasuprun@mail.ru; 3The Institute of Environmental and Agricultural Biology, University of Tyumen, 6 Volodarskogo St., 625003 Tyumen, Russia

**Keywords:** potato ring rot, detection, NASBA, Cas13a, one-pot assay

## Abstract

The ring rot of potato caused by the bacterial pathogen *Clavibacter sepedonicus* is a quarantine disease posing a threat to the potato industry worldwide. The sensitive and selective detection of *C. sepedonicus* is of a high importance for its effective control. Here, the detection system is reported to determine viable bacteria of *C. sepedonicus* in potato tubers, based on the coupling of CRISPR/Cas13a nuclease with NASBA (Nucleic Acid Sequence Based Amplification)—the method of isothermal amplification of RNA. Detection can be conducted using both instrumental and non-instrumental (visual inspection of test tubes under blue light) modes. When NASBA and Cas13a analyses were carried out in separate test tubes, the limit of detection (LOD) for the system was 1000 copies of purified target 16S rRNA per NASBA reaction or about 24 colony-forming units (CFUs) of *C. sepedonicus* per 1 g of tuber tissue. The testing can also be conducted in the “one-pot” format (a single test tube), though with lower sensitivity: LOD was 10,000 copies of target RNA or about 100 CFU per 1 g of tuber tissue for both instrumental and visual detection modes. The overall time of NASBA/Cas13a analysis did not exceed 2 h. The developed NASBA/Cas13a detection system has the potential to be employed as a routine test of *C. sepedonicus*, especially for on-site testing.

## 1. Introduction

Potato is ranked as the third most important food crop in terms of global consumption and recommended by the UN Food and Agriculture Organization as a food security crop [1]. Asia and Europe are leaders in potato production, accounting for 50% and 31%, respectively, of the total annual world production of about 380 million tons [2]. Bacterial diseases are significant biotic constraints of potato production, causing severe damage, especially in the tubers, the economically most important part of the plant [3].

*Clavibacter sepedonicus* (previously classified as *C. michiganensis* subsp. *sepedonicus* [4]) is a bacterial pathogen causing ring rot in potato—a quarantine disease threatening the potato industry around the globe [5]. Ring rot can lead to substantial direct (due to wilting and tuber rotting in field and store) and indirect (restrictions on cropping, expenses on disinfection and disposal, halt on the export) economic losses. The pathogen often remains undetected, especially at low incidence, and certification by visual inspection cannot provide the required level of disease control, so the asymptomatic infection can allow its spreading to regions free of the disease [5].

To date, a number of serological and PCR-based tests were developed to detect *C. sepedonicus* in symptomless potato tubers [6,7,8,9,10,11,12,13,14,15,16]. For example, the ELISA immunoassay based on monoclonal antibodies and approved in North America for diagnosis of *C. sepedonicus* enables the detection of 10^5^ to 10^6^ bacterial cells per mL [12]. In general, PCR-based methods were found to be more sensitive than serological methods, allowing detection down to a few copies of *C. sepedonicus* genomes per PCR reaction (or 10^3^–10^4^ bacterial cells per mL) [7,8,9,13]. TaqMan quantitative PCR was proposed as a primary screening test in EPPO standard methods [14]. Alongside the laboratory tests, the assays for on-site detection of *C. sepedonicus*, relying on the use of isothermal amplification techniques such as LAMP (loop-mediated isothermal amplification) or NASBA (nucleic acid sequence-based amplification), were suggested [15,16], thus making the mass primary screening for pathogen presence feasible in non-laboratory settings. However, the LAMP assay developed in [15] was unable to detect less than 50 pg of *C. sepedonicus* genomic DNA (or about 1000 bacterial genomes) per reaction. In the case of NASBA-based detection [16], the region in the 16S rRNA of *C. sepedonicus* was targeted and amplified by NASBA—the isothermal RNA amplification resulting from the concurrent actions of three enzymes, viz. AMV reverse transcriptase, RNase H, and T7 RNA polymerase. The generated RNA amplicons were determined with a molecular beacon—the method originally suggested by Leone et al. and known as AmpliDet [17]—providing a limit of detection (LOD) of about 10,000 copies of 16S rRNA (or one *C. sepedonicus* cell) per reaction [16], which even exceeded the sensitivity of PCR-based methods.

In general, isothermal amplifications, including NASBA, still lag behind PCR in regard to sensitivity and selectivity, mostly due to nonspecific amplification [18]. Since 2016, CRISPR/Cas nucleases (CRISPR, clustered regularly interspaced short palindromic repeats; Cas, CRISPR-associated protein), widely known as a successful toolkit for genomic editing [19], have been in focus as an effective solution to the problems associated with isothermal amplification [20]. Combining isothermal amplification with an RNA-guided CRISPR/Cas nuclease allows selective targeting of specific amplicons. Furthermore, the collateral activity acquired by a CRISPR/Cas nuclease after the binding of a guide RNA (gRNA) to a nucleic acid target can be utilized for signal enhancement. Indeed, when a specific segment of the gRNA sequence known as a “spacer” binds to a complementary sequence in a target (“protospacer”), the nonspecific cleavage of the substantial number of molecular reporters—short oligonucleotides labeled with a fluorophore and a fluorescence quencher (FQ reporters)—by each activated nuclease leads to a great amplification of the detection signal [21]. Today, there exists a steadily growing interest in CRISPR/Cas-based diagnostic systems, especially due to their advantages for on-site detection, including that of plant pathogens [22,23]. Among CRISPR/Cas nucleases utilized in such diagnostic systems, Cas13a nuclease, able to recognize single-stranded RNA targets, renders itself to be coupled with NASBA. However, few examples of such coupling related to virus or bacterial detection have been reported to date [24,25,26,27].

In this study, the NASBA/Cas13a detection system was developed to selectively detect viable cells of *C. sepedonicus* in tuber extracts with sensitivity exceeding that of the PCR-based methods and the NASBA-based AmpliDet method. The system was adapted for detection in the format of a single test tube (“one-pot” format), using both instrumental and non-instrumental (by a visual inspection of the test tube under blue light) detection modes to make it suitable for on-site testing of *C. sepedonicus*.

## 2. Results

### 2.1. Primers and gRNA Selection

To develop the NASBA/Cas13a assay for the detection of *C. sepedonicus*, regions in the bacterial 16S rRNA that were previously selected for detection with the AmpliDet method by Beckhoven et al. [16] were targeted. First, the artificial RNA target 40 nucleotides (nts) in length was synthesized (Appendix A). This target represented the section of NASBA amplicons corresponding to nucleotides from positions 116 to 155 in the alignment of the 16S rRNA genes for the *Clavibacter* species in [16] (the sequence of the section of the *C. sepedonicus* 16S rRNA gene, coding the section of the 16S rRNA amplified by NASBA is provided in Appendix A). This region differs by three to four single-nucleotide substitutions between *C. sepedonicus* and other *Clavibacter* species. Second, two variants of gRNA were designed (Appendix A) with gRNA spacers complementary to slightly different segments of the artificial RNA target. The gRNA variants were tested for their ability to recognize the artificial target when in complex with Cas13a and to produce the most effective cleavage of FQ reporters by the activated nuclease. Among these variants, gRNA1 demonstrated a higher efficiency, as judged by the initial rate of fluorescence increase, V_0_ (a slope of the linear segment of kinetic curves, Appendix A). Consequently, gRNA1 has been chosen for further experiments.

In NASBA, one of the primers, P1, has to include the sequence of the T7 promoter. Though it appears preferable to use the 20 nt long core sequence of the T7 promotor (taatacgactcactataggg) to minimize the overall primer length (thus decreasing the unwanted potential intra- and intermolecular base paring in primers), the use of simply the core sequence very rarely results in an acceptable amplification fold in NASBA. Various modifications of the T7 promoter have been suggested to boost the yield of RNA synthesized on a DNA template by T7 polymerase. The modifications usually consist of the addition of purine-rich sequences upstream and downstream of the core sequence [28,29]. These purine-rich sequences apparently stabilize the T7 polymerase binding to the promoter [30]. Since Beckhoven et al. reported no full sequence for the P1 primer, merely indicating the sequence which directly anneals to the corresponding segment of the 16S rRNA target [16], a number of P1 variants with various modifications of the T7 promoter were examined (Appendix A). The analysis of NASBA products by gel electrophoresis revealed that, as can be expected, no amplicons are generated if P1 contains only the core sequence of the T7 promoter (Appendix A). Two variants of P1 primers, viz. P1-218 and P1-222 (Appendix A), resulted in the occurrence of NASBA products whose positions on the gel reasonably agreed with the expected length of RNA amplicons (213, 215, 218, and 222 nts for P1-213, P1-215, P1-218, and P1-222, respectively; Appendix A). Next, the generated NASBA products were tested with a complex of Cas13a nuclease with gRNA1 by adding a 5 µL aliquot of the completed NASBA reaction to the 45 µL Cas13a/gRNA1 reaction mixture. Figure 1 demonstrates that NASBA carried out with the primer P1-222 provides the best result in terms of the Cas13a activity in regard to both the initial rate and the extent of fluorescence increase via the cleavage of FQ reporters. This is likely due to the lesser production of non-specific RNA amplicons compared to the primer P1-218 (Appendix A). As expected, no Cas13a activation was observed for NASBA products where no specific amplicons were revealed by gel electrophoresis (NASBA with primers P1-213 and P1-215, Appendix A). Finally, the gRNA and primers whose compositions are presented in Table 1 were chosen for the *C. sepedonicus* detection with the NASBA/Cas13a assay. It should be noted that the primer P1-222 is also extended by four nucleotides complementary to the target sequence compared to the sequence of the original P1 primer used in the study of Beckhoven et al. [16].

### 2.2. Sensitivity and Selectivity of NASBA/Cas13a Detection System

As a first step in evaluating the designed NASBA/Cas13a detection system, the sensitivity of *C. sepedonicus* detection in the “two-test-tube” format was assessed using the instrumental measurement of fluorescence. Total RNA extracted from the cultured bacterial cells was serially 10-fold diluted and used as a template in NASBA reactions conducted for 90 min (the duration time commonly recommended for NASBA). After that, 5 µL of each reaction mixture was added to 45 µL Cas13a/gRNA1 reaction mixtures and fluorescence was measured in time (the representative kinetic curves and V_0_ values are shown in Figure 2). The amount of 0.01 pg of total RNA per NASBA reaction (that corresponds to about 10^3^ copies of 16S rRNA per reaction according to [16]) was consistently detected, with the mean V_0_ value exceeding 11-fold that for the Cas13a/gRNA1 reaction mixtures with the added aliquots of no-template controls for NASBA (Figure 2b). The similar LOD was obtained when testing was conducted in the presence of background RNA—1 ng of potato total RNA per NASBA reaction (Appendix A).

Next, the LOD was determined for *C. sepedonicus* detection in experimentally contaminated potato tuber extracts. The tuber samples were contaminated right prior to the tissue homogenization by spiking them with small aliquots of serially diluted suspensions of cultured bacteria in distilled water, followed by RNA extraction. The bacterial dilutions were immediately plated on agar to determine the colony-forming unit (CFU) numbers. Figure 3 shows the representative kinetic curves and V_0_ values for the contaminated tuber samples. The amount of 0.12 CFU per NASBA reaction (about 1200 copies of 16S rRNA) was reliably detected and corresponded to about 24 CFUs in 1 g of tuber tissue (the relevant calculations of LOD in the number of CFUs per 1 g of tuber tissue can be found in the Section 4.7). At that load, the values of V_0_ steadily exceeded by seven-fold on average those obtained for no-template controls of NASBA (Figure 3b).

To evaluate the selectivity of the NASBA/Cas13a detection system, alongside 2 strains of *C. sepedonicus*, 5 other *Clavibacter* species, 13 taxonomically-related species, and 2 taxonomically unrelated species were tested. The results are provided in Table 2 as the ratio of V_0_ defined for these species to V_0_ defined for the *C. sepedonicus* stain Ac-2753. The V_0_ values were derived from the kinetics of fluorescence increase after adding a 5 µL aliquot of the completed NASBA reaction (carried out with 100 pg of bacterial RNA that corresponds to ~10^7^ copies of 16S rRNA per reaction) to the 45 µL Cas13a/gRNA1 reaction mixture. The *C. sepedonicus* strains demonstrated the similar increase in fluorescence—the mean V_0_ values differed by 6% (Table 2). For all other strains tested, the rate of fluorescence increase was less by a factor of 7 to 30 compared to that for the *C. sepedonicus* strains. For no-target controls of NASBA, the V_0_ ratios fall in the range of 0.01 to 0.03.

### 2.3. The “One-Pot” NASBA/Cas13a Detection System

To conduct *C. sepedonicus* detection in the format of a single test tube (“one-pot” assay), the NASBA reaction and the FQ reporter cleavage by the activated Cas13a/gRNA1 complex were carried out in same reaction tube. For that, the 10 µL Cas13a/gRNA1 reaction mixture was pre-added inside the lid of the test tube containing the 10 µL NASBA reaction mixture and amplification was conducted at 38 °C. After the completion of NASBA, the pre-added volume was mixed with the amplification assay by hand shaking or a short spin in a bench mini-centrifuge and the increase in fluorescence accompanied the cleavage of FQ reporters (also performed at 38 °C), was recorded.

First, the NASBA reaction time was optimized to produce the number of RNA amplicons in 10 µL reaction mixture, sufficient to activate the Cas13a nuclease in the “one-pot” setup. As seen from Figure 4, the NASBA conducted for 45 to 60 min generates the amount of target amplicons, which appears enough for activating Cas13a so to cleave FQ reporters to approximately the same extent. Similar results were obtained down to the target concentration of 10^3^ copies of 16S rRNA per 10 µL NASBA reaction (Appendix A). No sufficient increase in fluorescence was consistently observed below that target concentration. Thus, the amplification time of 60 min was chosen for further experiments to ensure that the sufficient number of RNA amplicons is generated to effectively activate Cas13a, on one hand, and, on the other, that the incubation of Cas13a at 38 °C is not too long to affect the ability of Cas13a to acquire collateral activity. However, only for the target concentration of 10^4^ copies, the increase of fluorescence reliably differed from that for the no-template control (Appendix A). This sets the LOD for detection of RNA extracted from the pure bacterial culture with the “one-pot” NASBA/Cas13a detection system as 10^4^ copies of target per reaction.

To determine sensitivity of the “one-pot” NASBA/Cas13a assay for *C. sepedonicus* detection in potato tubers, the samples of RNA were utilized, which were derived from the experimentally contaminated tuber extracts and earlier used to determine the corresponding LOD in the “two-test-tube” format. Figure 5a shows the representative curves for fluorescence kinetics. The reliable increase in fluorescence was consistently observed for the load of 0.5 CFU per reaction (Figure 5a,b), thus setting the LOD as about 100 CFUs per 1 g of potato tuber tissue.

### 2.4. “Naked-Eye” Detection of the Test Results

The non-instrumental detection appears useful if testing has to be conducted in non-laboratory settings. To explore the feasibility of non-instrumental detection of *C. sepedonicus* with the designed NASBA/Cas13a assay, the test tubes were visually examined after the completion of cleavage reaction by placing them on a blue light transilluminator. However, at the FQ reporter concentration of 0.125 µM, used for instrumental detection, all test tubes turned out to exhibit fluorescence output too low to discern the differences. To optimize the assay for visual detection, the FQ reporter concentration was varied from 0.125 µM to 0.75 µM (Appendix A). The FQ reporter concentration of 0.75 µM appeared to provide the reaction output suitable for “naked-eye” detection. With that concentration of FQ reporters, loads as low as 10^4^ copies of 16S rRNA per reaction (in the case of purified bacterial RNA, Figure 6a) or 0.5 CFU per reaction (in the case of RNA isolated from the experimentally contaminated tuber extract, Figure 6b) can be visually detected providing LOD values similar to those determined for the instrumental detection in the “one-pot” setting (Appendix A and Figure 5). To make the judgment more objective, the reaction tubes can be photographed with a smartphone and the images treated with the free image analysis software ImageJ v.1.54g (https://imagej.net/ij/ (accessed on 30 August 2024)). The results of the image analysis as mean values for the density of fluorescence intensity (within a rectangular area inside the image of the reaction mixture, as illustrated in Appendix A) are shown in Figure 6c,d. As seen, the values for loads of 10^4^ copies of 16S rRNA per reaction or 0.5 CFU per reaction are statistically significantly different (*p* = 0.95) from the values for the no-template controls.

## 3. Discussion

The effective detection of *C. sepedonicus* at the symptomless stage of infection is undoubtedly of importance for preventing its spread. The early diagnostics of ring rot is also important for the successful use of newly emerging antibacterial compounds against *C. sepedonicus* [31,32]. Serological tests are a simple and quick approach to *C. sepedonicus* detection (especially those in a format of lateral flow immunoassay [10]) but are characterized by rather relatively low sensitivity, with LODs of 10^4^–10^7^ cells per mL [11]. Presently, the PCR-based methods appear to dominate the molecular diagnostics of ring rot, providing detection sensitivity at the level of a few to several tens of *C. sepedonicus* genomes per reaction (e.g., [7,8,9]). The NASBA/Cas13a detection system developed in the study is able to detect as few as 10,000 copies of 16S rRNA per reaction in the “one-pot” format and 1000 copies in the “two-test-tube” format. Considering that the number of 16S rRNA molecules per *C. sepedonicus* cell was estimated as ~10,000 copies [16], such sensitivity corresponds to one or fewer bacterial cells per reaction, which notably exceeds that of PCR-based methods for *C. sepedonicus* detection.

In terms of viable bacteria, the LOD of 2 to 20 CFUs per reaction has been reported for PCR detection [7]. It should be noted that the LOD in [7] was estimated by plotting the threshold cycle values against the number of CFUs per PCR test tube. In general, PCR analysis cannot distinguish between viable and non-viable cells. The application of PCR to *C. sepedonicus* detection relies on amplification of particular regions of genomic DNA [7,8,9,13,14] which can be preserved in dead cells for a relatively long period of time. The direct amplification of RNA with such method as NASBA are commonly considered as a convenient way to assess viability of cells since RNA is degraded in dead cells very quickly [33]. As to viable *C. sepedonicus* bacteria, the developed NASBA/Cas13a detection system demonstrated a good match between LODs in terms of copies of RNA targets calculated on the basis of a load of purified bacterial RNA per NASBA reaction and on the basis of the CFU number—1000 and 1200 copies (0.12 CFU), respectively. The reasonable agreement holds also for the “one-pot” format—LODs were 10,000 and about 5000 copies (0.5 CFU) per reaction. This indicates that the developed system does detect merely viable bacteria, even if RNA was isolated from the experimentally contaminated tuber extracts. Interestingly, the AmpliDet method was able to detect as low as 10,000 molecules of purified 16S rRNA per reaction but only as low as 100 CFUs per reaction were detected when RNA was extracted from the tubers contaminated with *C. sepedonicus* [16]. It is likely that the co-amplification of some potato RNA sequences, which can be present in a great excess in RNA samples derived from potato extracts, may affect the NASBA performance through competition for reaction reagents. Also, the possible presence of other bacteria should be taken into consideration. Compared to the P1 primer used by Beckhoven et al. [16], the P1 primer used in the designed NASBA/Cas13a detection system was extended by four nucleotides. The extended primer appears to anneal to the target sequence more specifically, as was justified by the results of the electrophoretic analysis of the NASBA products. Additionally, it is thought that in the NASBA/Cas13a detection system, the activated Cas13a nucleases can process the substantial amount of FQ reporters, thus providing a much greater level of fluorescence signal than the AmpliDet fluorescent probes for the same amount of specific NASBA amplicons. 

As demonstrated by Beckhoven et al., by testing 17 different strains of *C. sepedonicus* [16], sequences of *C. sepedonicus* 16S rRNA, which we have also chosen for targeting with the NASBA/Cas13a detection system, are conserved among the *C. sepedonicus* strains. The result for the two strains of *C. sepedonicus* which were tested in this study supports such a conclusion. As for the non-target species, the obtained results supplement the results of Beckhoven et al. [16], confirming the specificity of these sequences for *C. sepedonicus* (the 5 other *Clavibacter* species, 13 taxonomically-related species, and 2 taxonomically-unrelated species tested demonstrated drastically different fluorescence responses) and suggesting the high selectivity of the designed NASBA/Cas13a detection system, not inferior to that of PCR-based methods.

The sensitivity of the NASBA/Cas13a detection system in the “one-pot” and “two-test-tube” formats significantly differed: 10,000 vs. 1000 copies of purified 16S rRNA (or 0.5 CFUs vs. 0.12 CFUs) per reaction, respectively. It appears most likely that the 1 h long incubation of Cas13a inside the tube lid at 38 °C prior to mixing with the NASBA reaction affects the Cas13a performance. However, testing in a format of a single test tube would allow us to avoid potential aerosol contaminations, which can lead to false-positive results.

In summary, the NASBA/Cas13a-based assay was developed for selective detection of *C. sepedonicus* in potato tuber extracts. The assay can detect as few as 10^3^ copies of purified target RNA per NASBA reaction or 24 CFUs of *C. sepedonicus* per 1 g of tuber tissue (0.12 CFU per NASBA reaction) in the “two-test-tube” mode. The assay can also be carried out in a single test tube (“one-pot” format) with a LOD of 10^4^ copies of purified target RNA per reaction or 100 CFUs of *C. sepedonicus* per 1 g of tuber tissue (0.5 CFU per reaction). The sensitivity of the NASBA/Cas13a-based assay in both detection modes either exceeds or is at least not inferior to the sensitivity of PCR-based methods, while its overall duration (≤2 h) is comparable with that of PCR analysis. Furthermore, the detection can be conducted by assessing the reaction output with the naked eye, which makes the assay additionally suitable for on-site detection of the bacterium.

## 4. Materials and Methods

### 4.1. DNA and RNA Oligonucleotide Synthesis

DNA oligonucleotides (templates for enzymatic RNA synthesis and NASBA primers) were chemically synthesized and purified by Lumiprobe RUS Ltd. (Moscow, Russia). Their sequences are provided in Appendix A. The FQ reporters (FAM-5′-uuuuu-3′-BHQ1) were also synthesized and purified by Lumiprobe RUS Ltd. The gRNAs and the artificial RNA target were synthesized enzymatically with “TranscriptAid T7 High Yield Transcription Kit” (Thermo Fisher Scientific, Waltham, MA, USA). For that, the corresponding DNA oligonucleotides were equimolarily mixed with T7F oligonucleotide (Appendix A) and annealed by heating and cooling to form DNA templates for RNA synthesis. The synthesized RNAs were purified with a mixture of phenol/chloroform/isoamyl alcohol 25:24:1 (Acros Organics, Geel, Belgium), followed by ethanol precipitation. The precipitates were dissolved in nuclease-free water, and RNA concentrations were determined using a NanoDrop 1000 spectrophotometer (Thermo Fisher Scientific). The quality of RNA was assessed by denaturing polyacrylamide gel electrophoresis. Aliquots of RNA solutions were stored at −80 °C until further use.

### 4.2. Bacteria Culturing and RNA Isolation

All bacteria, except for *Clavibacter* species, were cultivated at 28 °C (in the case of *E. coli*—at 37 °C) in LB medium [34] received from Dia-M (Moscow, Russia). The *Clavibacter* species were grown at 28 °C on agar medium with the following composition: 20 g/L of agar, 10 g/L of casein–peptone, 5 g/L of yeast extract (all from Becton Dickinson, Franklin Lakes, NJ, USA), 5 g/L of glucose (Fluka, Buchs, Switzerland), 5 g/L of NaCl (Merck, Darmstadt, Germany), pH 7.0–7.2. To prepare suspensions of *Clavibacter* species, bacteria were harvested from the agar plates and resuspended in sterile distilled water. To measure the number of CFUs, the aliquots of the serially diluted suspensions of the *C. sepedonicus* strain Ac-2753 were applied to agar plates. The number of visible colonies (CFUs) present on an agar plate was counted, multiplied by the dilution factor, and corrected by the volume of the added aliquots to obtain the CFU concentration in the initial suspension.

Total RNA was extracted from bacteria pelleted by centrifugation with the spin column-based “Total RNA Isolation Mini Kit” (Agilent, Santa Clara, CA, USA). Following the manufacturer’s suggestion for RNA extraction from Gram-positive bacteria, the bacterial pellet was first resuspended in 100 µL of the lysis solution (LS) containing 3 g/L of lysozyme (Merck) dissolved in TE buffer (10 mM Tris-HCl, 1 mM EDTA, pH 8.0) and the suspension was incubated at ambient temperature for 30 min. The further steps of RNA extraction were carried out according to manufacturer’s protocol for the spin column-based purification of RNA. The extraction of total RNA from the tuber tissue, both contaminated and uncontaminated with *C. sepedonicus*, was performed using the spin column-based “SKYSuper Plant Genomic DNA” isolation kit (SkyGen, Moscow, Russia) with the following modification. The initial amount of potato tuber tissue taken into analysis was increased up to 2 g but only 5% of this amount was, in fact, used for the actual extraction of nucleic acids on the spin column (to match the load per a column recommended by the manufacture). The increase in the initial amount of tissue taken for homogenization allowed us to extend down the range of CFU values per 1 g of tissue when determining the LOD. First, the piece of potato tuber was cut into smaller pieces and placed (together with an aliquot of the *C. sepedonicus* dilution where required) into a mortar and liquid nitrogen was added. The tuber tissue pieces were thoroughly grinded with a pestle. Once the liquid nitrogen evaporated, 2 mL of LS was added, the powder was resuspended, the suspension was placed into a 5 mL nuclease-free Eppendorf tube, and the volume was brought up to 5 mL with LS. Afterwards, the 250 µL aliquot was withdrawn and used for RNA extraction following the manufacturer’s protocol, except that the step with the RNase treatment of tissue homogenate was omitted and the additional step of DNase treatment using the “RNase-Free DNase Set” from Qiagen (Hilden, Germany) was introduced. For that, 20 µL of the DNase solution was placed onto a spin column membrane with bound nucleic acids after the first washing steps and the column was left on a bench for 15 min at ambient temperature. The DNase solution was removed by centrifugation and the column membrane was washed twice in accordance with the manufacturer’s protocol, followed by RNA elution. The RNA concentrations in the eluate were determined on a Qubit 4.0 fluorimeter with the “Qubit RNA BR Assay Kit” (Thermo Fisher Scientific). The extracted RNA was brought to a desirable concentration with nuclease-free water, aliquoted, and stored at −80 °C.

### 4.3. NASBA

To perform RNA amplification, the lyophilized NASBA enzyme mixtures were employed from the commercial NASBA kit manufactured by AmpliSence (Moscow, Russia). The kit is aimed at determining *Chlamydia trachomatis* in clinical samples and contains tubes with lyophilized enzyme mixture (AMV reverse transcriptase, RNase H, and T7 RNA polymerase), tubes with lyophilized nucleotides, two rehydration buffers (buffer 1 and 2), and tubes with a set of particular primers and a probe specific for *Chlamydia trachomatis*. Tubes with the primers and probe were discarded and primers targeting *C. sepedonicus* 16S rRNA were used instead. The NASBA reaction mixtures were prepared following the manufacturer’s instruction. The lyophilized nucleotides were dissolved with 86 µL of rehydration buffer 1, and then 16 µL of the primer mixture (containing 5 µM of each primer, Appendix A and Table 1) and 8 µL of nuclease-free water were added. The solution was divided into 10 aliquots and placed in PCR tubes. Then, 2 to 4 µL of a particular total RNA sample (or nuclease-free water for no-template control) was added to each tube. The reaction mixtures were brought up to a final volume of 15 µL with nuclease-free water where necessary, followed by incubation at 65 °C for 5 min and then at 41 °C for 4 min. Meanwhile, the lyophilized enzyme mixture was dissolved in 50 µL of rehydration buffer 2 and incubated for 15 min at ambient temperature. Afterwards, 5 µL of the dissolved enzyme mixture was added to each PCR tube containing the solution of nucleotides, primers, and RNA target (where required), and the tubes were incubated for 90 min at 41 °C.

### 4.4. Cas13a Cleavage Assay

The recombinant *Leptotrichia buccalis* CRISPR/Cas13a nuclease was utilized to design the Cas13a cleavage assay. The nuclease was expressed using the p2CT-His-MBP-Lbu_C2c2_WT plasmid vector received from the Addgene repository (www.addgene.org (accessed on 20 May 2024)). The expression and purification were performed similar to that in [35]. For the preparation of Cas13a used, the optimal nuclease concentration and Cas13a/gRNA molar ratio were determined with the artificial RNA target (Appendix A) as described in [36]. Finally, the 40 µL Cas13a/gRNA mixture contained 15 nM of Cas13a, 60 nM of gRNA, and 1 unit/µL of murine RNase inhibitor (Merck) in the reaction buffer (RB) composed of 50 mM Tris-HCl (pH 7.3), 75 mM NaCl, and 7.5 mM MgCl_2_. The mixture was incubated at 37 °C for 15 min, and, afterwards, 5 µL of FQ reporters with concentration of 1.25 µM in nuclease-free water and 5 µL of a NASBA sample (or a sample of artificial RNA target) were added. For negative controls, 5 µL of nuclease-free water were added instead of RNA-containing samples. The reaction mixture was immediately placed into a well of a microplate if measurements were to be conducted on an Infinite M200 PRO plate reader (TECAN, Mannedorf, Switzerland) at 37 °C. Otherwise, a PCR tube with the reaction mixture was placed into the heating block of a DTprime*5* thermal cycler (DNA-TECHNOLOGY, Moscow, Russia) set at the constant temperature of 37 °C. The fluorescence was measured every 30 to 60 s for 10 min. For each time point of measurements, the values of fluorescence for the negative control were subtracted from those obtained with NASBA samples.

### 4.5. The “One-Pot” Assay

To detect *C. sepedonicus* in a format of a single test tube, the NASBA reaction was prepared similar to that as described in Section 4.3 and 10 µL was placed into the bottom of a PCR tube. Immediately, 10 µL of the Cas13a reaction mixture prepared to contain 30 nM of Cas13a, 120 nM of gRNA, 2 unit/µL of murine RNase inhibitor (Merck), and 0.25 to 1.5 µM of FQ reporters in RB was placed inside the lid of the PCR tube. The tube was transferred into the heating block of a DTprime5 thermal cycler and set at the constant temperature of 38 °C. The fluorescence was measured every 60 s for 110–120 min. In 15 to 60 min after the incubation started, the tube was withdrawn and the Cas13a reaction mixture was blended with the NASBA reaction by hand shaking or a short centrifugation. The tube was returned to the heating block and the measurements continued. For the “naked-eye” detection, the tubes were placed on a Dark Reader transilluminator (Clare Chemical Research, Dolores, CO, USA). The illuminated tubes were observed and photographed through an orange filter supplied with the transilluminator.

### 4.6. Statistical Treatment

To characterize performance of the CRISPR/Cas13a detection system in the two-tube or the single-tube formats, the arithmetic means of V_0_ values or values of the density of fluorescence intensity determined by ImageJ (v.1.54g) analysis were calculated based on results of 3 to 5 independent experiments. The corresponding confidence intervals were calculated using the two-tailed Student’s *t*-distribution (https://mathworld.wolfram.com/Studentst-Distribution.html (accessed on 5 September 2024)) for the confidence level *p* = 0.95.

### 4.7. The Calculation of LOD in Colony-Forming Units per 1 g of Tuber Tissue

The number of CFUs in 2 g of the experimentally contaminated potato tuber, *N*, was determined by plating serial bacterial dilutions on agar medium as described in Section 4.2 for *C. sepedonicus* cultivation. Since after the disruption of bacterial and plant cells with liquid nitrogen, only 5% of RNA of the added bacteria were taken for RNA isolation (250 µL out of 5 mL of the tuber tissue homogenate; see Section 4.2 for details), *N* has to be divided by 20. Further, only 5% of the isolated total RNA were loaded into a NASBA or NASBA/Cas13a reaction tube. Thus, the number of CFUs per reaction was calculated as follows: *N*/20/20 = *N*/400. When the limit of detection in CFUs per reaction tube (LOD_t_) was experimentally determined, it was used to calculate the LOD in CFUs per 1 g of tuber tissue as LOD_t_ · 400/2 g = LOD_t_ · 200.

## Figures and Tables

**Figure 1 ijms-25-12218-f001:**
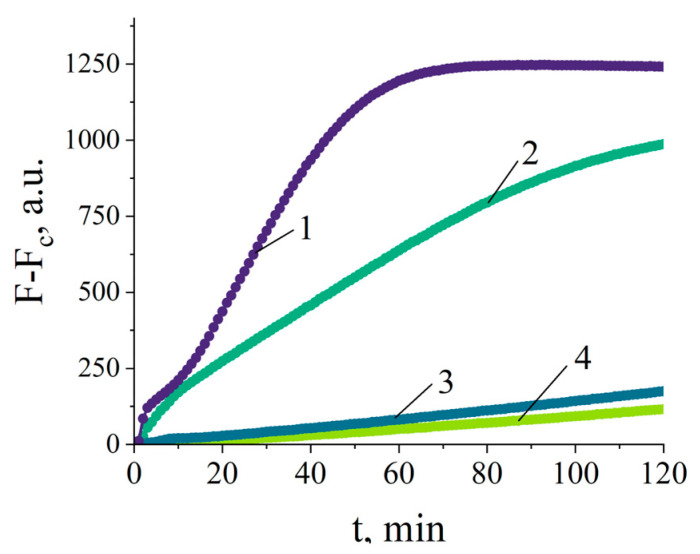
The representative curves of fluorescence kinetics due to FQ cleavage by Cas13a nuclease carried out for NASBA reaction products. Curves 1, 2, 3, and 4—NASBA with primers P1-222, P1-218, P1-213, and P1-215, respectively. F and F_c_ are values of fluorescence with aliquots of NASBA reactions conducted in the presence (10^5^ copies of 16S rRNA per NASBA reaction) or absence (no template control of NASBA) of the RNA target, respectively. Measurements of fluorescence were performed on an Infinite M200 PRO plate reader. a.u.—arbitrary units of fluorescence.

**Figure 2 ijms-25-12218-f002:**
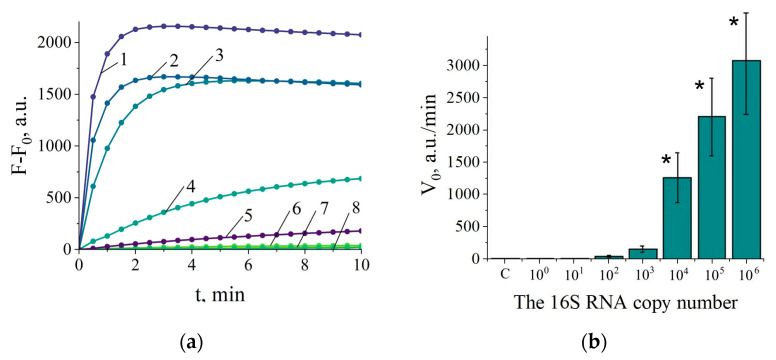
The performance of the NASBA/Cas13a detection system in the “two-test-tube” format. (**a**) The representative fluorescence curves. F and F_0_—fluorescence measured in Cas13a analysis of NASBA samples and that of an aliquot of nuclease-free sample, respectively. a.u.—arbitrary units of fluorescence. Curves 1, 2, 3, 4, 5, 6, and 7—10^6^, 10^5^, 10^4^, 10^3^, 10^2^, 10, and 1 copy of 16S rRNA per reaction, respectively. Curve 8—with no template control of NASBA. (**b**) The V_0_ values in the Cas13a cleavage assay for NASBA samples with different loads of RNA target. “C” corresponds to the analysis of no-template controls for NASBA. All measurements were made in triplicate. The mean values and confidence intervals for the confidence level *p* = 0.95 are shown. Bars representing statistically significant differences from that for no-template control are marked with asterisk.

**Figure 3 ijms-25-12218-f003:**
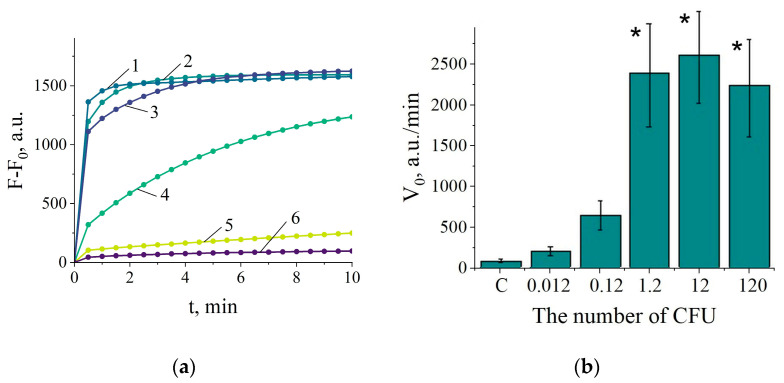
Determination of *C. sepedonicus* CFUs in the experimentally contaminated potato tuber extracts with the NASBA/Cas13a detection system in the “two-test-tube” format. (**a**) The representative kinetic curves illustrating the Cas13a analysis of NASBA products. F and F_0_ are as in Figure 2. Curves 1, 2, 3, 4, and 5—120, 12, 1.2, 0.12, and 0.012 CFUs per NASBA reaction, respectively. Curve 6—with no-template control for NASBA. (**b**) The V_0_ values in the Cas13a cleavage assay for NASBA samples with different CFU loads. “C” corresponds to the analysis of no template controls for NASBA. The mean values for 5 replicates and the corresponding confidence intervals for *p* = 0.95 are shown. Bars representing statistically significant differences from that for no template control are marked with asterisk.

**Figure 4 ijms-25-12218-f004:**
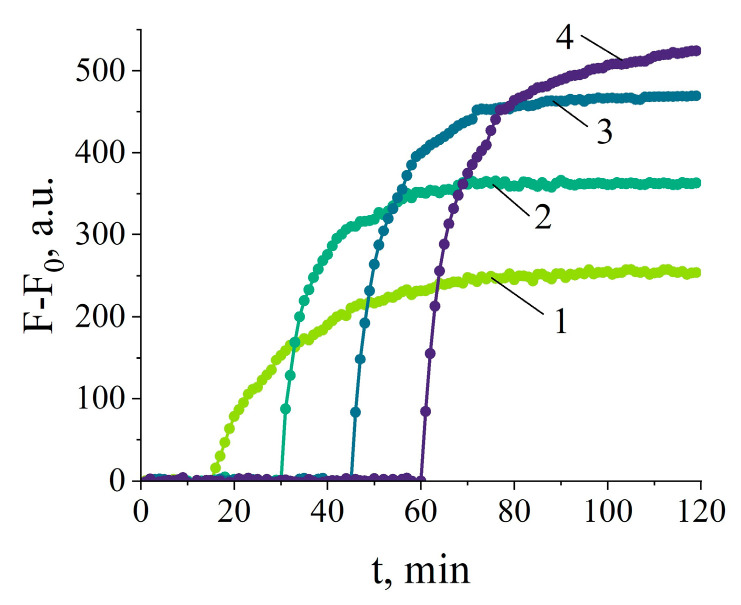
The representative curves of fluorescence kinetics due to FQ cleavage by Cas13a nuclease, carried out in the single-test-tube format (the “one-pot” assay). F_0_—the fluorescence intensity at the start of measurements. Curves 1, 2, 3, and 4—NASBA conducted for 15, 30, 45, and 60 min, respectively, followed by mixing the 10 µL NASBA reaction with the 10 µL Cas13a cleavage assay mixture by a short spin of the reaction tube. The number of target RNA in NASBA—10^5^ copies per reaction.

**Figure 5 ijms-25-12218-f005:**
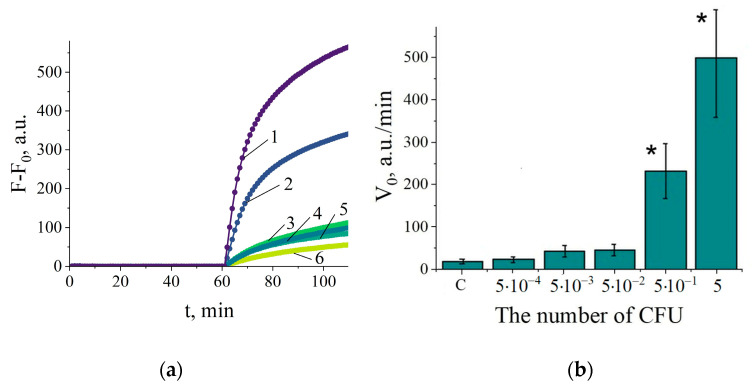
Determination of *C. sepedonicus* in potato tuber extracts with the NASBA/Cas13a system in the single-test-tube format. (**a**) The representative kinetic curves. F_0_—the fluorescence intensity at the start of measurements. Curves 1, 2, 3, 4, and 5—5, 0.5, 0.05, 0.005, and 0.0005 CFUs per reaction, respectively. Curve 6—no template control. (**b**) The V_0_ values for different loads of CFUs per reaction. “C”—no-template control. The mean values for 5 replicates and the corresponding confidence intervals for *p* = 0.95 are shown. Bars representing statistically significant differences from that for no-template control are marked with asterisk.

**Figure 6 ijms-25-12218-f006:**
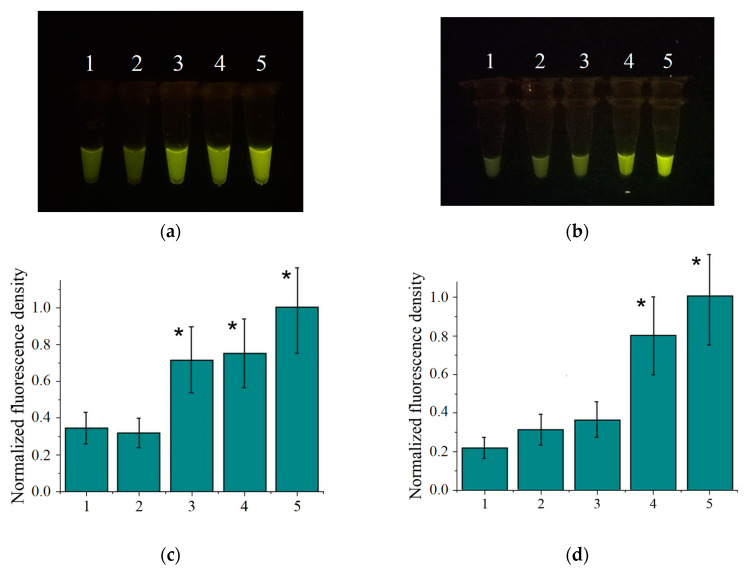
Determination of *C. sepedonicus* in the single-test-tube format with “naked-eye” detection. (**a**) The representative images of test tubes for the purified bacterial RNA. 1—no-template control; 2, 3, 4, and 5—10^3^, 10^4^, 10^5^, and 10^6^ copies of 16S rRNA per reaction, respectively. (**b**) The representative images of test tubes for RNA isolated from the experimentally contaminated tuber extracts. 1—no-template control; 2, 3, 4, and 5—0.005, 0.05, 0.5, and 5 CFUs per reaction, respectively. (**c**,**d**) The values of normalized fluorescence density for *C. sepedonicus* detection using the purified bacterial RNA and RNA isolated from the experimentally contaminated tuber extracts, respectively. Bar numbering in bar graphs corresponds to that in panels (**a**,**b**). The normalization is carried out by dividing the fluorescence density values by the maximal one. All measurements were performed in triplicate. The mean values and confidence intervals for *p* = 0.95 are shown. Bars representing statistically significant differences from that for no-template control are marked with asterisk.

**Table 1 ijms-25-12218-t001:** Sequences of NASBA primers (P1-222 and P2) and gRNA selected for the NASBA/Cas13a detection system. The sequence of the extended T7 promoter is shown in italics, and the sequence of the gRNA spacer is underlined.

Oligonucleotide Name	Oligonucleotide Sequence (5′ → 3′)
P1-222	*aattctaatacgactcactatagggagaagggg*ttggccccggcagtctccta
P2	cgatgcaacgcgaagaac
gRNA1	gggauuuagaccaccccaaaaaugaaggggacuaaaacagaaacgugcagagaugugcgccccccaa

**Table 2 ijms-25-12218-t002:** The list of the strains used in the study. The strain Ac-2753 of *C. sepedonicus* was used as a reference to normalize V_0_ values.

Specie	Host	Origin	Collection ^1^	Strain Number	V_0_ Ratio ^2^
* Clavibacter sepedonicus *	potato	USA	VKM	Ac-2753	1.00 ± 0.07
* Clavibacter sepedonicus *	potato	USA	VKM	Ac-1405	0.94 ± 0.08
* Clavibacter michiganensis *	tomato	Zambia	VKM	Ac-1144	0.05 ± 0.01
* Clavibacter michiganensis *	tomato	USA	VKM	Ac-1403	0.04 ± 0.02
* Clavibacter phaseoli *	common beans	Spain	VKM	Ac-2641	0.13 ± 0.02
* Clavibacter insidiosus *	alfalfa	USA	VKM	Ac-1402T	0.05 ± 0.01
* Clavibacter nebraskensis *	maize	USA	VKM	Ac-1404T	0.11 ± 0.02
* Clavibacter tesselarius *	wheat	USA	VKM	Ac-1406T	0.05 ± 0.02
* Dickeya zeae *	maize	USA	DSMZ	DSM 18068	0.04 ± 0.01
* Dickeya chrysonthemi *	Chrysanthemum morifolium	USA	DSMZ	DSM 4610	0.05 ± 0.02
* Dickeya paradisiaca *	Musa paradisiaca	Colombia	IPO	2127	0.09 ± 0.01
* Dickeya dadantii *	Pelargonium capitatum	Comoros	DSMZ	DSM 18020	0.04 ± 0.02
* Dickeya dianthicola *	Dianthus caryophyllus	United Kingdom	DSMZ	DSM 18054	0.03 ± 0.01
* Dickeya solani *	potato	Russia	VNIIF	1C3	0.04 ± 0.01
* Pectobacterium versatile *	potato	Russia	VKM	B-3416	0.05 ± 0.02
* Pectobacterium aquaticum *	potato	Russia	VKM	B-3417	0.04 ± 0.02
* Pectobacterium polaris *	potato	Russia	VKM	B-3420	0.03 ± 0.01
* Pectobacterium parmentieri *	potato	Russia	VKM	B-3423	0.05 ± 0.02
* Pectobacterium carotovorum *	potato	Denmark	VKM	B-1247	0.03 ± 0.02
* Pectobacterium brasiliensis *	potato	Russia	VKM	B-3424	0.03 ± 0.01
* Pectobacterium brasiliensis *	potato	Russia	VKM	B-3425	0.04 ± 0.02
* Pectobacterium odoriferum *	potato	Russia	VNIIF	1557	0.03 ± 0.01
* Agrobacterium tumefaciens *	undefined	USA	VKM	B-1573	0.07 ± 0.01
* Escherichia coli *	clinical isolate	USA	VKM	B-3034	0.03 ± 0.01

^1^ VKM—All-Russian Collection of Microorganisms; DSMZ—German Collection of Microorganisms and Cell Cultures; IPO—Collection of the Institute for Phytopathological Research, the Netherlands; VNIIF—Collection of the All-Russian Scientific Research Institute of Phytopathology. ^2^ The mean values and the standard errors for three replicates are shown.

## Data Availability

Data is contained within the article and Appendix A.

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
