# Peer review of "Detection of Potato Pathogen Clavibacter sepedonicus by CRISPR/Cas13a Analysis of NASBA Amplicons"

_ijms, 2024, doi:10.3390/ijms252212218_

Round 1
Reviewer 1 Report
Comments and Suggestions for Authors
The manuscript reports a methods developed to detect low rate of infection of an important quarantine pathogen of potato. The authors should reorganize some parts of the manuscript, allocating them to the appropriate section and must eliminate the use of personalization in the whole manuscript. Some of the words used should be replaced by appropriate terminology and some concept must be explained better or revised. An annotated version of the manuscript is provided for detailed revision.

Moderate English revision and elimination of personalization must be done.
Author Response
- We are very grateful to the Reviewer for critical comments and suggestions which were highly useful in improving the manuscript. All changes recommended by the Reviewer have been implemented in the manuscript except for providing conclusions as a separate section and a change of the manuscript title. The IJMS instruction for authors states that the conclusions section “is not mandatory but can be added to the manuscript if the discussion is unusually long or complex”. We don not feel that it is our case and would prefer to make a summary at the end of the discussion. We would also like to leave the mentioning of host name in the article title. The reason is that one of Reviewers asked to make a separate paragraph in the Introduction, devoted to the importance of potato, its cultivation area, and problems faced by its cultivation. In our opinion, it would be better to leave "potato" in the title for logic of presentation.
- The pdf file of the manuscript with the Reviewer's comments and recommendations and our responses to each point raised by the Reviewer is provided as an attachment. All changes made in the manuscript upon revision are highlighted with yellow.

Reviewer 2 Report
Comments and Suggestions for Authors
The manuscript, titled "Detection of potato pathogen Clavibacter sepedonicus by CRISPR/Cas13a analysis of NASBA amplicons," presents the integration of the CRISPR/Cas13a nuclease with the NASBA technique for the identification of viable C. sepedonicus bacteria in potato tubers. The study suggests that the NASBA/Cas13a detection system developed holds promise as a routine diagnostic tool for C. sepedonicus, particularly suitable for on-site testing. The objective of this paper is commendable; however, it requires resolution of several significant issues prior to acceptance.
1. The introduction should provide a more comprehensive discussion on the limitations of current detection methods, outlining how the present study addresses these challenges.
2. Please specify the number of biological repeats conducted for each experiment. This information should be clearly stated in the methods section.
3. The manuscript introduces a "one-pot" detection mode; however, it lacks a direct comparison with traditional detection methods. It is crucial to include a comparative analysis of the sensitivity and specificity of the "one-pot" method versus conventional approaches to demonstrate the advantages or disadvantages of the new method.
4. Lines 119-125, Lines 153-162, Lines 175-183, Lines 226-231, Lines 240-248, Lines 271-280: the paper should describe the biological repeat times in methods and figure legends of Figures 1-6.
Comments on the Quality of English LanguageModerate editing of English language required.
Author Response
We are thankful to the Reviewer for useful recommendations which were helpful in improving the manuscript.
Comment 1: The introduction should provide a more comprehensive discussion on the limitations of current detection methods, outlining how the present study addresses these challenges.
Response 1: The introduction has been rewritten to address the Reviewer’s recommendation, including providing additional references (refs. 10-12 of the revised manuscript). The limitations of the current detection methods were stressed (lines 47-52, 57-59, and 64-66 of the revised manuscript). The goal of the study was presented in a more clear way (lines 86-91).
Comment 2: Please specify the number of biological repeats conducted for each experiment. This information should be clearly stated in the methods section.
Response 2: The number of repetitions has been provided in the method section of the initial version of the manuscript in the subsection 4.6 “Statistical treatment”. Upon revision, the number of repeats is also provided in figure legends to address the Reviewer suggestion.
Comment 3: The manuscript introduces a "one-pot" detection mode; however, it lacks a direct comparison with traditional detection methods. It is crucial to include a comparative analysis of the sensitivity and specificity of the "one-pot" method versus conventional approaches to demonstrate the advantages or disadvantages of the new method.
Response 3: As a response to Reviewer criticism, we have re-arranged the discussion section to provide a direct comparison of sensitivity of the developed detection method with that of traditional methods (lines 296-308, 322-326 of the revised manuscript). Three additional references (refs. 10-12) are given. Serological methods are known to be less sensitive that the PCR-based ones. The PCR-based methods are characterized by LODs of a few of C. sepedonicus genomes (or bacterial cells) per PCR reaction, at best (refs. 7-9 of the revised manuscript). The developed method allows for detecting as low as 10,000 copies of 16S rRNA per reaction in the “one-pot” format and 1,000 copies in the “two test tubes” format. Since C. sepedonicus cell is estimated to contain 10,000 copies of 16S rRNA at average, the sensitivity of detection with the NASBA/Cas13a system corresponds to one or less bacterial cell per reaction that exceeds that of PCR-based methods. As to selectivity of detection, it is hardly possible to make a direct comparison, in rigorous terms. We demonstrated with a number of non-target species that the detection is selective (discussed on lines 348-353 of the revised manuscript). However, we cannot demonstrate that with the same strains used to confirm selectivity for the PCR-based methods.
Comment 4: Lines 119-125, Lines 153-162, Lines 175-183, Lines 226-231, Lines 240-248, Lines 271-280: the paper should describe the biological repeat times in methods and figure legends of Figures 1-6.
Response 4: The number of repetitions was provided in the figure legends upon revision as recommended by the Reviewer. In the method section, the number of repetitions was given in the initial version of the manuscript in the subsection 4.6 “Statistical treatment”.
Reviewer 3 Report
Comments and Suggestions for Authors
A method with coupling of 16 CRISPR/Cas13a nuclease and NASBA was developed to determine C. sepedonicus in potato tubers. It provides a rapid and convenient way to monitor the pathogen in potato. The whole design and organization is good. And the work labor is enough. However, minor revision on format or writing should be addressed.
1. line 41, citation expression. line 88, the content in citation should be given. line 288.
2. Figure 1, confirm the unit in y-axis. Fig 2 and 3, the significance mark should be added on column in (b). Figure 6 c and d, the same problem.
3. line 390, the species should be italic, please double-check through the article.
Author Response
We thank the Reviewer for positive evaluation of our work and suggestions which were useful in improving the manuscript.
Comment 1: line 41, citation expression. line 88, the content in citation should be given. line 288.
Response 1: Line 41 (line 47 in the revised manuscript) – the citation was corrected upon revision, all relevant references are given; line 88 (line 99 in the revised manuscript) – the content in citation is provided as new Table S2 of the Supplementary information file (as now stated on lines 99-101 of the revised manuscript); line 288 – unfortunately, we could not figure out what the Reviewer wanted us to address on this line, nothing on that line gave us a clue.
Comment 2: Figure 1, confirm the unit in y-axis. Fig 2 and 3, the significance mark should be added on column in (b). Figure 6 c and d, the same problem.
Response 2: The explanation of “a.u.” (arbitrary units of fluorescence) has been provided upon revision in legends for Figs. 1 and 2. The significance marks have been added in Figs. 2, 3, 5, and 6 on columns presenting fluorescence or initial rate values statistically different from those of no template controls.
Comment 3: line 390, the species should be italic, please double-check through the article.
Response 3: The species names were double-checked throughout the manuscript and corrected where necessary.
Reviewer 4 Report
Comments and Suggestions for Authors
Please see attachment

Author Response
â–ª Thank you so much for the manuscript. The work is excellent, beneficial, and important. It is well-written and organized.
We thank the Reviewer for appraising our work.
â–ª I noticed that there are many phrases with the same author, why did the authors not merge them or at least they should write the same author/authors mentioned that
We do not quite understand what changes the Reviewer want us to make. We assumed that the Reviewer means the reference to the paper by Beckhoven et al. (ref. 16 in the revised manuscript, previously ref. 7). We cited this work many times through the manuscript because the study we undertook is closely related to and, to some extent, originated from that work. We find it necessary to highlight its importance for our study when presenting and discussing our results, by referring to that work specifically (as Beckhoven et al. instead of simply giving its number in the list of references). And we would prefer to leave it this way unless the Reviewer is insisting on otherwise. If we misunderstood what the Reviewer meant by this comment, we kindly ask for clarification.
â–ª The figures in this manuscript are clear and well organized but the titles are long if the authors can summarize them, it will be good
We agree with the Reviewer that figure legends turned to be quite lengthy. As the response to the Reviewer concern, we have shortened legends for Figs. 1, 2, 3, and 5 to same extent, but fail to do it for Figs. 4 and 6. In our opinion, the further shortening of figure legends will lead to a loss of substantial information necessary to understand the figures. If the Reviewer does not insist, we would prefer to leave the figure legends as they are now. Otherwise, we would have to move the essential information to the main text that could be inconvenient for a reader.
â–ª In figures 6 (a and b) are not clear, the authors should improve them, for me they are not visible.
Figs. 6, a and b, were improved upon revision to address the Reviewer concern.
â–ª The title of table 2 is so long, it should be summarized to be more precise and informative.
The title of table 2 was shortened upon revision. Some technical details were moved to the main text (lines 199-203, 209-210).
â–ª Many sections were written without references such as statistical analysis and the paragraph entitled “The calculation of LOD in colony-forming units per 1 g of tuber tissue”. The authors should write references to support their work.
The two-tailed Student’s t-test employed for statistical analysis is a widely used statistics described in numerous textbooks. This is why we did not provide any specific reference to that test in the initial version of the manuscript. The reference to the two-tailed Student’s t-test is given upon revision in the manuscript main text as a link to a relevant website (lines 486-487) to address the Reviewer recommendation. The subsection 4.7 “The calculation of LOD in colony-forming units per 1 g of tuber tissue” simply describes in details how we converted the experimentally defined CFU number in 2 g of potato tissue into the number of CFUs per reaction, taking into account all dilutions and proportions between volumes of solutions used. In our opinion, no references are necessary to support these calculations.
â–ª The manuscript on potato crops, in my opinion, the authors should write a separate paragraph on the importance, cultivated area, and the problems faced by its cultivation.
The separate paragraph on potato importance, cultivation area, and the problem faced by its cultivation has been incorporated into the Introduction section to address the Reviewer recommendation (lines 32-37).
â–ª Clarify the NASBA and Cas13a components
The components of NASBA are provided in more details (namely, the composition of the enzyme mixture) on lines 430-431 upon revision. Unfortunately, the concentrations of enzymes and dNTP, as well as compositions of rehydration buffers 1 and 2 are not disclosed by the manufacturer. It should be noted that their concentrations and buffer compositions are rather those commonly employed in other NASBA tests. As least, when we used another commercial NASBA kit, namely the liquid NASBA kit from AMSBIO (https://www.amsbio.com/nasba-kits-reagents), the results were even a bit better in regard to detection sensitivity. However, in that kit the NASBA components were not in a lyophilized form, so we preferred to switch to the lyophilized NASBA kit from the domestic company AmpliSence. The components of Cas13a reaction mixture and their concentrations are given on lines 454-460.
â–ª The author should describe the RNA purification step
For the RNA extraction from bacterial culture, the commercial kit from Agilent was used as indicated on lines 394. The kit utilizes spin columns to selectively bind bacterial RNA and all procedures are similar to those commonly used for the spin column-based nucleic acid purification. We specifically pointed out upon revision that the kit is the spin column-based kit (lines 393-394). For Gram-positive bacteria, the manufacturer recommends to treat the bacteria with the home-made solution of lysozyme to disrupt bacterial cell walls. This procedure was described in details on lines 394-398. As to the extraction RNA from potato tubers contaminated with C. sepedonicus, we employed the spin column-based kit aimed at extraction of DNA. In fact, both DNA and RNA can be bound to the membrane of the spin column. In the standard protocol, RNA is degraded by the pretreatment of tissue homogenate with RNase A prior to the column purification. We modified the procedure by omitting the RNases pretreatment and introducing the on-column DNase treatment as an additional step. As to the rest, we followed the manufacturer’s protocol which was similar to other protocols for the spin column-based nucleic acid purification. The pertinent details were added to the RNA extraction procedure upon revision to make the description clear (lines 403-408, 415-422).
â–ª One-pot format has lower sensitivity (100 CFU/g) compared to separate tubes (24 CFU/g)
Indeed, the developed NASBA/Cas13a detection system had lower sensitivity in the “one-pot” format compared to that in the “two test tubes” format. It is most likely that the 1 h long incubation of Cas13a inside the tube lid at 38oC prior to mixing with NASBA reaction affects the Cas13a performance. We discussed this issue in the revised manuscript (lines 348-353) and are thankful to the Reviewer for drawing our attention to this point.
â–ª No statistical analysis was shown in the figures. How can we justify the significance of the experiment
The significant differences were marked in the figures with asterisk to address the Reviewer concern.
Round 2
Reviewer 2 Report
Comments and Suggestions for Authors
The current paper has been essentially revised and can be accepted.
Comments on the Quality of English LanguageMinor editing of English language required.
Reviewer 4 Report
Comments and Suggestions for Authors
Thank you so much for your efforts in improving your manuscript